# Translation, Adaptation, and Validity of the Short Food Literacy Questionnaire for Brazil

**DOI:** 10.3390/foods11243968

**Published:** 2022-12-08

**Authors:** Larissa Baungartner Zeminian, Ligiana Pires Corona, Isabelle do Nascimento Batista, Marcela Chagas da Silva, Diogo Thimoteo da Cunha

**Affiliations:** 1Laboratório Multidisciplinar em Alimentos e Saúde, Faculdade de Ciências Aplicadas, Universidade Estadual de Campinas—UNICAMP, Rua Pedro Zaccaria, 1300, Limeira 13484-350, Brazil; 2Laboratório de Epidemiologia Nutricional, Faculdade de Ciências Aplicadas, Universidade Estadual de Campinas—UNICAMP, Rua Pedro Zaccaria, 1300, Limeira 13484-350, Brazil

**Keywords:** diet, food, and nutrition, feeding behavior, food preferences, knowledge

## Abstract

Food literacy refers to the knowledge and skills related to healthy food choices. The aim of this study is to present the process of the cross-cultural adaptation and content validation of the Short Food Literacy Questionnaire (SFLQ) for the Brazilian population. The process of adaptation and validation of the SFLQ was conducted in four steps: (1) conceptual and item equivalence; (2) translation with semantic equivalence; (3) operational equivalence; (4) measurement equivalence. The study involved ten judges (food and nutrition experts) and 20 subjects (general population) in evaluating the semantic equivalence of the SFLQ. The semantic equivalence was assessed by calculating the content validity coefficient (CVC). All the items had a CVC greater than 0.80 considering all the evaluated parameters. The SFLQ has been adapted for the Brazilian context and guidelines. The adapted version of the questionnaire was referred to as the SFLQ-Br. The operational equivalence was conducted with 120 subjects using an online approach. The measurement equivalence was assessed using confirmatory factor analysis, a reliability assessment, and an instrument stability assessment. A single factor was extracted, and all the items had a factor loading of >0.40 and appropriate goodness-of-fit values. All the equivalence results show that the SFLQ-Br can be used in the Brazilian population as a reliable, consistent, and stable instrument to measure food literacy.

## 1. Introduction

Food literacy (FL) refers to the knowledge and skills related to healthy food choices, including knowledge of how foods are grown and practices related to their selection, preparation, and consumption [1]. FL expresses the eating behaviors necessary to plan, manage, select, prepare, and consume food and can be used in various contexts, including public, research, and policy [2]. FL is a focus, or specific domain, of health literacy in the food context [3]. Health literacy is defined by Nutbeam (1998) as “the cognitive and social skills which determine the motivation and ability of individuals to gain access to, understand, and use information in ways which promote and maintain good health.” [4] p. 357. In this sense, health and FL go beyond objective knowledge to include the motivation and competence to understand, appraise, and apply health information [5].

The definitions from FL include components from six key themes: Skills and Behaviors, Food/Health Choices, Culture, Knowledge, Emotions, and Food Systems. That is, the term FL is applied to both cognitive (knowledge) and functional (skills and behaviors) abilities, as well as to the individual (food/health choices) and the social level (culture, emotions, and food systems) [6]. FL is classified as a factor that contributes to health in general. A lack of data that measures and monitors the food environment’s comprehensive aspects shows that measuring and predicting food behavior is still challenging [7]. In Brazil, for example, no studies were found that addressed FL. Among the various factors that explain the limitation of research in this area may be the lack of an assessment tool suitable for FL in the Brazilian food context.

A systematic review on this topic found that there are only two instruments to assess FL in the adult population, one of which is the Short Food Literacy Questionnaire (SFLQ) [8]. The SFLQ was initially developed in Switzerland as part of an intervention study to reduce salt consumption among workers [9]. The instrument includes items such as: Access to and understanding of food and nutrition information, knowledge of the Swiss food pyramid, Swiss recommendations for fruit, vegetable, and salt consumption, as well as the ability to prepare healthy meals, the ability to support others in nutrition issues, the ability to select and evaluate relevant information, and the ability to assess the long-term health effects of their dietary habits. The questionnaire can be used to plan and evaluate public health interventions focusing on FL to improve understanding of the topic [10]. In addition, because it is a short instrument, SFLQ is easy to use in routine health care and research.

Improving the FL of the population contributes to healthier food choices, leading to better nutrition and health. In addition, families’ food knowledge and behaviors influence government food policy and regulatory decisions [11]. In Brazil, recent household surveys have shown that the consumption of fresh and low-processed foods has decreased, and the consumption of processed and ultra-processed foods has increased [12]. Despite several socioeconomic factors driving this consumption pattern, it is observed that Brazilian consumers have difficulty classifying foods. They attribute health to several ultra-processed foods with little or no health benefits [13]. FL is paramount to improving food behaviors since cognitive aspects are essential in these decisions [14].

As mentioned above, there is currently no validated tool in Brazil to assess FL. In this sense, the SFLQ can be translated and adapted for this purpose. Currently, SFLQ can be found in the original English version (published version) [10], the German version (original version), and the cross-cultural validation for Turkey [15]. Using an instrument created in other countries but culturally adapted allows the comparison of results and elaborating cross-cultural study models. Therefore, this study aims to present the process of the cross-cultural adaptation and content validity of the SFLQ for applicability to the Brazilian population.

## 2. Materials and Methods

### 2.1. The Scale

The cross-cultural adaptation and content validity of the SFLQ followed the universalistic approach, which involves the assessment of equivalence between the original instrument and the instrument being adapted. The adapted version is called SFLQ-Br in this work. The process of the SFLQ-Br adaptation and validation was organized into four steps: (1) conceptual and item equivalence; (2) semantic equivalence; (3) operational equivalence; (4) measurement equivalence [16,17]. Figure 1 depicts the study flow, including the sample and process of each step. The research was conducted from October 2021 to April 2022.

The SFLQ is a self-completion questionnaire developed by Krause et al. While the SFLQ was developed in the German language, it was published in English. Thus, we decided to adapt the published version [10]. The SFLQ consists of 12 items whose responses vary from 4 to 5 points on a Likert scale. The answers are given on the following scales: ‘1—disagree strongly’ to ‘4—agree strongly’; ‘1—very bad’ to ‘5—very good’; ‘1—very hard’ to ‘4—very easy’; and ‘1—never’ to ‘5—always’ [9,10].

The original authors of the SFLQ were consulted for approval of the validation study. The SFLQ is made available by the University of Bern under a Creative Commons: the Attribution-Noncommercial-Share Alike (CC-BY-NC-SA) license [18]. Therefore, the SFLQ and the SFLQ-Br are freely available and may be used for non-commercial purposes. Still, the original authors of both questionnaires must always be credited as the authors.

The State University of Campinas Ethics Committee approved the study (CAAE 53319421.3.0000.8124). All participants signed an informed consent form and were informed that they could withdraw from the study at any point.

### 2.2. Conceptual and Item Equivalence

Conceptual and item equivalence is to evaluate the relevance and appropriateness of the original instrument and the new context by exploring the construct of interest [16,17]. Thus, a literature review, discussion, and analysis of the topic and existing instruments for assessing FL were conducted. This analysis allowed us to identify the SFLQ as an interesting instrument for this purpose, with possibilities of adaptation for Brazil, keeping the original concept of the instrument.

### 2.3. Semantic Equivalence

Semantic equivalence corresponds to the validation of the questionnaire’s content to verify that the meaning of the concepts of the original version is maintained in the translated version of the instrument [16,17]. The following steps were performed: (a) translation by bilinguals, (b) semantic and theoretical evaluation by experts, (c) review of semantic evaluation by experts, and (d) evaluation of semantic equivalence between the original version and the revised version by people who have no affinity with the subject (general population).

For the translation of the original version into Portuguese, two bilinguals (Brazilians with fluency in English) analyzed the questionnaire independently. The two versions obtained were compared and evaluated by the researchers. Two researchers carried out the synthesis of the translations to produce a version that had conceptual equivalence and understandable language. Thus, version 1 of the SFLQ-Br was completed.

The first version of the questionnaire was presented to 10 judges [19]. They were food and nutrition professors with at least a Ph.D. degree and a specialization in nutrition, dietetics, or public health. The judges were selected from the analysis of the Lattes curriculum and invited to participate by e-mail. They received an Excel spreadsheet containing the original version and translation/adaptation of the 12 items of the SFLQ, as well as the fields to complete three indicators: Clarity of language (Is the item clear enough for the population? Is it easy to read and understand?); Practical Relevance (Do you think this item is relevant to the instrument?); and Theoretical Relevance (Does this item represent an important part of the food literacy assessment?). There was also a box to enter comments and suggestions. Responses to each indicator for each item were provided on a scale of 1 to 5 (‘1—not at all’ to ‘5—completely’).

The SFLQ-Br was analyzed by the Content Validity Coefficient (CVC) [20] in two consecutive rounds. The CVC of each item (CVCi) was calculated by dividing the average of the judges’ judgment scores by the maximum score of the last category of the Likert scale, in this case, ‘5’, and subtracting the standard error of polarization of the judges. In turn, the standard error of polarization was calculated by the ratio between 1 and the absolute number of judges and raised to the absolute number of judges. The items that presented a CVCi > 0.80 in the three indicators (CVC-Clarity, CVC-Practical Relevance, and CVC-Theoretical Relevance) were considered adequate [19].

Finally, individuals with no affinity for the topic (general population) assessed the semantic equivalence of the revised version. Twenty participants were selected, five from each age group, between 18–29 years, 30–45 years, 46–59 years, and over 60 years. All participants were recruited online through social media. The participants received an Excel spreadsheet with the 12 items of the questionnaire (questions and answers) to assess three indicators: linguistic clarity (is the language clear?), appropriateness (is the language appropriate for your age group?), and comprehensibility (did you clearly understand the question?). The responses were given on a scale of 1 to 5 (‘1—not at all’ to ‘5—completely’). There was also a yes or no fill-in box related to the need to change the item and a box for suggested changes. CVCs with values closer to one are better. Therefore, it was considered appropriate if the scale content validity (S-CVC) value was equal to or higher than 0.90 [19].

### 2.4. Operational Equivalence

The operational equivalence reflects the possibility of using a similar questionnaire format, instructions, method of administration (electronic format), and methods of measurement of the current scale in the target population [16,17].

The sample size was estimated using the N:p ratio of 10, i.e., at least 10 cases for each indicator or 12 in the case of SFLQ-Br [21]. The participants were invited unintentionally through social networks. The SFLQ-Br was made available via Google Forms. This step was conducted electronically with 120 individuals over the age of 18. The test-retest reliability was measured by reapplying the SFLQ-Br with 29 participants from the original sample.

### 2.5. Measurement Equivalence

Measurement equivalence consists of examining the psychometric properties of the final version of the instrument based on the following approaches: an assessment of the dimensional validity and adequacy of factors; an assessment of the reliability of information; an assessment of instrument stability [16,17].

The dimensionality found in previous studies was tested by confirmatory factor analysis (CFA) using diagonally weighted least squares. In previous studies, the SFLQ had a single factor with 12 items, with item #2 scored by the average. The chi-square value (*χ*^2^ with *p* < 0.05), root mean square error of approximation (RMSEA < 0.08), comparative fit index (CFI > 0.90), standardized root mean square residual (SRMR < 0.08), Tucker–Lewis index (TLI > 0.90), and goodness-of-fit index (GFI > 0.90) were used to check the model fit [22]. It was also checked whether the factor loading of each item was greater than 0.40.

The second approach targets the formal assessments of scale reliability (internal consistency and stability) to assess the extent to which an instrument’s scores are free of random errors, contributing to the procedural appropriateness of cross-cultural adaptation. The internal consistency was evaluated using the McDonald’s Omega (*ω* > 0.70). McDonald’s Omega is a similar measurement to Cronbach’s Alpha. However, the former indicator overcomes many limitations of the latter, e.g., Cronbach’s alpha underestimates the true reliability when the items are not tau-equivalent [23]. The correlation of the items was made by Pearson’s correlation. The CFA and internal consistency were conducted using JASP 0.16.4 (JASP Team-Netherlands-Open-source tool, supported by the University of Amsterdam). The stability was measured by the quadratic weighted kappa value considering the test–retest data. At least a moderate agreement (>0.40) was considered appropriate [24].

## 3. Results

Table 1 shows the original version of the SFLQ and two versions of the SFLQ-Br. The second version is the final version after the adjustments from the conceptual and semantic equivalence process.

The judges also evaluated the scales to assess each item of the SFLQ-Br (Table 2). The items and response scales were assessed together.

Researchers adapted several items to the Brazilian context in creating the 1st (translated) version of the SFLQ-Br. As with the redefinition of terms, those containing the “Swiss Food Pyramid” were replaced with the “Dietary Guidelines for the Brazilian Population (*Guia Alimentar para População Brasileira*)”, which is the official Brazilian document for dietary recommendations (questions 3, 4, and 5). The terms “healthy diet” and “healthy nutrition” were also replaced with “healthy eating (*alimentação saudável*)” for ease of understanding (questions 1, 8, and 11). The judges also proposed changes in the terminology to align it with the Dietary Guidelines for the Brazilian Population and the vocabulary used in Brazil.

Table 3 shows the average CVCi assigned by the judges.

Two questions were rated as unacceptable in the analysis of version 1 of the SFLQ-Br. Question #2 showed 0.78 on the CVC-Clarity, and question #7 showed 0.78 on the CVC-Theoretical-relevance. In this way, the researchers analyzed the comments and suggestions and changed these questions to create version 2 of the SFLQ-Br. Question 2 was dismembered because, according to the judges, it was unclear whether each item (a, b, c, d, or e) required a response or whether it was a general response that included an understanding of these types of nutrition information. The term “in the past” was removed from question #7 because of temporal bias (difficulty in remembering or assigning an appropriate timeline). In addition, this removal was necessary since no question refers to the present to compare and track the development of nutrition literacy. Questions #4 and #5 were also adjusted. Although all CVCs were acceptable, these questions were amended according to the Dietary Guidelines for the Brazilian Population to adapt them to the Brazilian context. Some other term adjustments were made for better understanding, such as replacing “judge (*julgar*)” with “evaluate (*avaliar*)” and “appropriate (*apropriado*)” with “adequate (*adequado*)”. After the adjustments, the second version of the SFLQ-Br was resent to the judges, and all modified items showed adequate CVCs.

The evaluation by the general public revealed that all the items of the questionnaire were rated as acceptable (CVCi > 0.80) in terms of linguistic clarity, appropriateness, and comprehensibility (Table 4). The S-CVC was 0.93. For all the questions, the percentage of those who did not want to make changes was over 80%, except for question #3 (75%). However, it was decided to keep it in the proposed format because the percentage was not too low and since all the CVCis of question #3 were considered acceptable. Thus, version 2 was validated by the judges, approved by the general population, and became the final version of the SFLQ-Br.

The 120 individuals who participated in the SFLQ-Br operational equivalence had an average age of 35 years old, and 75% were women. Table 5 shows the results of the CFA and test–retest. The CFA analysis showed a reasonable fit with the chi-square test (*x*^2^ = 142.68; *p* < 0.001), and RMSEA = 0.11, CFI = 0.98, SRMR = 0.10, TLI = 0.98, and GFI = 0.98. The factor loading of each SFLQ-Br item was greater than 0.40. The scale was found to be reliable and stable by internal consistency analysis (McDonald’s *ω* = 0.874) and stability (the kappa of all items > 0.50). All items showed significant correlations (*r* > 0.20; *p* < 0.05) with all other items, indicating a convergent validity (see the Appendix A for the correlations).

Based on the original methodology, a final SFLQ score can be calculated. This final score is the sum of the scores of the individual questions. For question 2, the average score is used. Based on this, the final score range is 8 to 50. We observed a final average score of 33.2 ± 7.9. The original method does not specify thresholds for food literacy scores.

## 4. Discussion

The study was designed to allow the application of the SFLQ in the Brazilian population since there is no validated instrument in Brazil to assess FL. The adaptation of an existing instrument has advantages over the elaboration of a new one, e.g., it is possible to compare data collected in different samples and contexts, which allows greater equity in the assessment. The use of adapted instruments allows a better generalization and a better study of the differences, similarities, and characteristics between populations [25]. Several countries, such as the United States [26], Italy [27], the Netherlands [28], and Taiwan [29], already have validated instruments to assess FL. In addition, as already mentioned, the SFLQ is available in three different languages. The SFLQ-Br, the first validated instrument to measure FL in the Brazilian population, is vital to understanding Brazilians’ nutrition knowledge and predicting dietary behaviors and motivators. The questionnaire can also be used to inform about government decisions and public policies [11].

During the assessment with the judges, the importance of including the Dietary Guidelines for the Brazilian Population [30] in the SFLQ-Br was noted. Although this adaptation changes the original meaning of the SFLQ in some ways (especially in questions #4 and #5), the research instruments must represent local guidelines and recommendations. The Dietary Guidelines for the Brazilian Population is recognized as an innovative guideline by including the degree of food processing [31]. As a result, all items of the SFLQ-Br had a CVCi above 0.80 for the judges and the general population. Also, the S-CVC was higher than 0.90. Using the SFLQ-Br, it was found that question #3 on the knowledge of the Dietary Guidelines for the Brazilian Population had an average of less than 3.0. This result indicates that the population considers their knowledge of the guidelines to be relatively low, which highlights the importance of including the guidelines in the tools, instruments, and interventions for the population.

The factor structure of the SFLQ-Br was found to have a reasonable fit, with good estimates found for the several assumed parameters. The scale was found to be acceptable with the fit values obtained by the CFA, with all the items having a factor loading of >0.40, i.e., all the items actively contribute to the construct of “food literacy.” Items 4, 5, 9, and 10 of the SFLQ-Br were characterized by the highest factor loadings, indicating that these items made a greater contribution to the factor and maintained the representativeness of the original variables [32].

The indicators CFI, TLI, and GFI showed values above 0.95. These results show that the variance and covariance structure of the data matrix provides a very good fit with the proposed model [33]. However, the RMSEA and SRMR values were close to the 0.10 limit, with the SRMR having a slightly better fit value [33]. These latter two indicators are more difficult to be discussed since no study using the SFLQ presented such indicators, making it impossible to compare them. For instruments with ordinal scales, the SRMR seems to be a better option to assess the model’s fit [34]. The single-factor structure is the one proposed in the original instrument [10] and is confirmed by a high-reliability value observed for this single factor.

Moderate (>0.40) to near-perfect (>0.80) values of weighted kappa were also observed [24], indicating good stability of the questionnaire. Stability is the extent to which similar scores are obtained at two different time points, indicating the consistency of repeated measures, i.e., how stable the measure is over time [35].

Therefore, considering all the indicators measured and the lack of studies showing such indicators when using the SFLQ, we consider that the SLFQ-Br presented a satisfactory fit. The combination of several indices must be regarded as a criterion for classifying the model as appropriate or not [22]. Therefore, considering all the presented indicators, the SFLQ-Br can be considered reliable, consistent, and stable. However, it is necessary to reevaluate the SRMR and RMSEA in the CFA in new studies in Brazil to better discuss these indicators.

One of the limitations of the SFLQ and, consequently, of the SFLQ-Br, is the lack of criteria to classify food literacy. In this sense, it is important for researchers and practitioners to understand that the higher the score, the better the food literacy. Therefore, food literacy can be used as an explanatory variable and for measures of association and can be followed up in longitudinal studies or practical interventions. Another limitation was the lack of back-translation. While many studies use only forward-translation, the back-translation might be helpful for minor adjustments.

Finally, the SFLQ-Br is ready and available for use in research, service, primary care, and anywhere where the measure of FL in adults is relevant. Further testing is required for use in populations under 18 years of age. Improving the FL of the population is necessary and urgent. Ultra-processed food intake has increased in Brazil [12]. This increase is partly due to cognitive problems, such as a lack of knowledge about the risks associated with ultra-processed foods [13], a preference for palatable foods [36,37], and poor FL.

## 5. Conclusions

After comparing the original version of the SFLQ with version 2 of the SFLQ-Br, it was found that the adaptations of the questionnaire items are equivalent and did not change their original cultural meaning. Moreover, the different indicators guarantee the instrument’s reliability, validity, and stability. It was found that the Brazilian version of the SFLQ is adequate. Therefore, the SFLQ-Br can be considered as a valuable instrument for assessing Brazilian adults’ FL.

## Figures and Tables

**Figure 1 foods-11-03968-f001:**
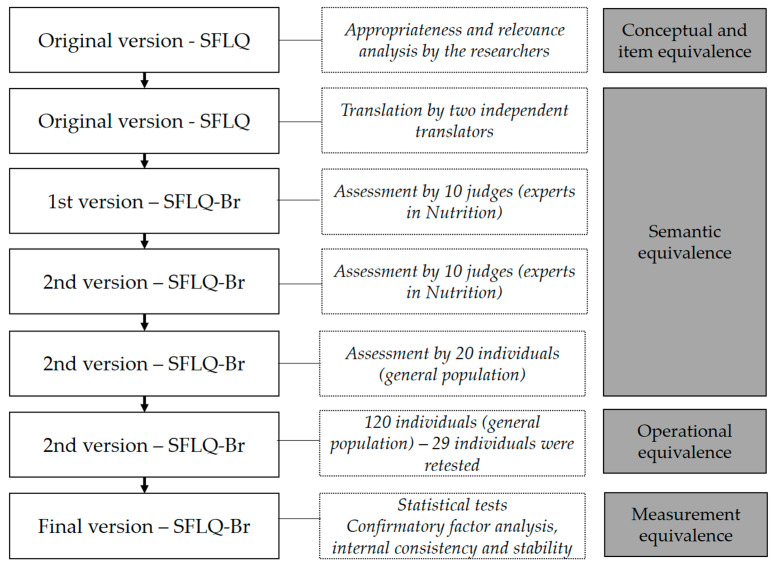
Study flow, including the sample and process of each step.

**Table 1 foods-11-03968-t001:** Items of the original version of the SFLQ and the 1st version and 2nd version of the SFLQ-Br.

Item	Original Version	1st Version *	2nd Version *
#1	When I have questions on healthy nutrition, I know where I can find information on this issue	*Quando eu tenho dúvidas sobre alimentação saudável, eu sei onde eu posso encontrar informações sobre este assunto.*	*Quando eu tenho dúvidas sobre alimentação saudável, eu sei onde eu posso encontrar informações sobre este assunto.*
#2	In general, how well do you understand the following types of nutritional information?	*Em geral, quão bem você entende os seguintes tipos de informação nutricional?*	*Em geral, o quanto você entende os seguintes tipos de informação sobre nutrição?*
#2a	(A) Nutrition information leaflets	*(A) Folhetos com informações nutricionais*	*(A) Folhetos/folderes com recomendações nutricionais*
#2b	(B) Food label information	*(B) Rótulos nutricionais nos alimentos*	*(B) Rótulos nutricionais nos alimentos*
#2c	(C) TV or radio program on nutrition	*(C) Programas de TV ou Rádio sobre nutrição*	*(C) Programas de TV ou Rádio sobre nutrição*
#2d	(D) Oral recommendations regarding nutrition from professionals	*(D) Recomendações verbais de profissionais relativas à nutrição*	*(D) Recomendações verbais de profissionais sobre nutrição*
#2e	(E) Nutrition advice from family members or friends	*(E) Conselhos de membros da minha família ou de amigos sobre nutrição*	*(E) Orientações de membros da minha família ou de amigos sobre nutrição*
#3	How familiar are you with the Swiss Food Pyramid?	*Quão familiarizado(a) você está com o Guia Alimentar para População Brasileira?*	*Quão familiarizado(a) você está com o Guia Alimentar para População Brasileira?*
#4	I know the official Swiss recommendations about fruit and vegetable consumption	*Eu conheço as recomendações oficiais do Guia Alimentar para População Brasileira sobre o consumo de frutas e vegetais.*	*Eu conheço as recomendações oficiais do Guia Alimentar para População Brasileira sobre o consumo de alimentos in natura e minimamente processados.*
#5	I know the official Swiss recommendations about salt intake.	*Eu conheço as recomendações oficiais do Guia Alimentar para População Brasileira sobre o consumo de sal.*	*Eu conheço as recomendações oficiais do Guia Alimentar para População Brasileira sobre óleos, gorduras, sal e açúcar.*
#6	Think about a usual day: how easy or difficult is it for you to compose a balanced meal at home?	*Pensando em um dia normal: quão fácil ou difícil é para você montar uma refeição balanceada em casa?*	*Pensando em um dia normal: quão fácil é para você montar/preparar uma refeição saudável em casa?*
#7	In the past, how often were you able to help your family members or a friend if they had questions concerning nutritional issues?	*No passado, com que frequência você foi capaz de ajudar os membros de sua família ou um(a) amigo(a) caso eles tivessem dúvidas relativas a assuntos nutricionais?*	*Com que frequência você é capaz de ajudar os membros de sua família ou um(a) amigo(a) caso eles tenham dúvidas relativas à nutrição?*
#8	There is a lot of information available on healthy nutrition today. How well do you manage to choose the information relevant to you?	*Há muitas informações disponíveis sobre alimentação saudável atualmente. Quão bem você consegue selecionar as informações relevantes para você?*	*Há muitas informações disponíveis sobre alimentação saudável atualmente. Quão bem você consegue selecionar as informações relevantes para você?*
#9	How easy is it for you to judge if media information on nutritional issues can be trusted?	*Quão fácil é para você julgar se informações da mídia sobre questões nutricionais são confiáveis?*	*Quão fácil é para você avaliar se informações da mídia sobre nutrição são confiáveis?*
#10	Commercials often relate foods with health. How easy is it for you to judge if the presented associations are appropriate or not?	*Comerciais frequentemente associam alimentos à saúde. Quão fácil é para você julgar se as associações apresentadas são apropriadas ou não?*	*Propagandas frequentemente associam alimentos à saúde. Quão fácil é para você avaliar se as associações apresentadas são adequadas ou não?*
#11	How easy is it for you to evaluate if a specific food is relevant for a healthy diet?	*Quão fácil é para você avaliar se um alimento específico é relevante para uma alimentação saudável?*	*Quão fácil é para você avaliar se um alimento específico é relevante para uma alimentação saudável?*
#12	How easy is it for you to evaluate the longer-term impact of your dietary habits on your health?	*Quão fácil é para você avaliar o impacto a longo prazo dos seus hábitos alimentares na sua saúde?*	*Quão fácil é para você avaliar o impacto a longo prazo dos seus hábitos alimentares na sua saúde?*

* Presented in Brazilian Portuguese.

**Table 2 foods-11-03968-t002:** Response scales of each item of the original version of the SFLQ and the 1st version and 2nd version of the SFLQ-Br.

Item	Original Version	1st Version *	2nd Version *
#1	1-Disagree strongly = 1 to Agree strongly = 4; I do not have experience with these issues = 0	*Discordo totalmente = 1 a Concordo totalmente = 4; Não possuo experiência no assunto = 0*	*Discordo totalmente = 1 a Concordo totalmente = 4; Não possuo experiência no assunto = 0*
#2	Very bad = 1 to Very good = 5; I do not make use of this kind of information = 0	*Muito mal = 1 a Muito bem = 5; Não faço uso desse tipo de informação = 0*	*Muito mal = 1 a Muito bem = 5; Não faço uso desse tipo de informação = 0*
#3	Very bad = 1 to Very good = 5	*Muito mal = 1 a Muito bem = 5*	*Muito mal = 1 a Muito bem = 5*
#4	Disagree strongly = 1 to Agree strongly = 4	*Discordo totalmente = 1 a Concordo totalmente = 4*	*Discordo totalmente = 1 a Concordo totalmente = 4*
#5	Disagree strongly = 1 to Agree strongly = 4	*Discordo totalmente = 1 a Concordo totalmente = 4*	*Discordo totalmente = 1 a Concordo totalmente = 4*
#6	Very hard = 1 to very easy = 4; not applicable = 0	*Muito difícil = 1 a Muito fácil = 4; Não se aplica = 0*	*Muito difícil = 1 a Muito fácil = 4; Não se aplica = 0*
#7	1 = Never to always = 5; there have never been any questions = 0	*Nunca = 1 a Sempre = 5; Nunca houve nenhuma dúvida = 0*	*Nunca = 1 a Sempre = 5; Nunca houve nenhuma dúvida = 0*
#8	Very bad = 1 to Very good = 5; I have not been interested in these issues = 0	*Muito mal = 1 a Muito bem = 5; Eu não sou interessado(a) nestes assuntos = 0*	*Muito mal = 1 a Muito bem = 5; Eu não sou interessado(a) nestes assuntos = 0*
#9	Very difficult = 1 to very easy = 4	*Muito difícil = 1 a Muito fácil = 4*	*Muito difícil = 1 a Muito fácil = 4*
#10	Very hard = 1 to very easy = 4	*Muito difícil = 1 a Muito fácil = 4*	*Muito difícil = 1 a Muito fácil = 4*
#11	Very hard = 1 to very easy = 4	*Muito difícil = 1 a Muito fácil = 4*	*Muito difícil = 1 a Muito fácil = 4*
#12	Very hard = 1 to very easy = 4	*Muito difícil = 1 a Muito fácil = 4*	*Muito difícil = 1 a Muito fácil = 4*

* Presented in Brazilian Portuguese.

**Table 3 foods-11-03968-t003:** Average CVCi of the SFLQ-Br based on judge’s rates.

Item	1st Version SFLQ-Br			2nd Version SFLQ-Br		
	Clarity	Practical Relevance	Theoretical Relevance	Clarity	Practical Relevance	Theoretical Relevance
#1	0.90	0.98	1.00	-	-	-
#2	**0.78**	0.84	0.90	0.96	0.98	0.98
#2a	-	-	-	0.90	0.96	0.96
#2b	-	-	-	0.98	1.00	1.00
#2c	-	-	-	0.94	0.96	0.92
#2d	-	-	-	0.98	1.00	1.00
#2e	-	-	-	0.96	0.92	0.84
#3	0.86	1.00	0.98	-	-	-
#4	0.94	0.82	0.88	-	-	-
#5	0.98	0.88	0.88	-	-	-
#6	0.80	0.86	0.86	-	-	-
#7	0.90	0.80	**0.78**	0.92	0.92	0.88
#8	0.88	0.94	0.92	-	-	-
#9	0.82	0.94	0.96	-	-	-
#10	0.82	0.98	1.00	-	-	-
#11	0.90	0.92	0.94	-	-	-
#12	0.90	1.00	1.00	-	-	-
Average CVC	0.87	0.91	0.92	-	-	-

Bold values = values below the 0.80 threshold; dashed cells = not evaluated.

**Table 4 foods-11-03968-t004:** CVC of the 2nd version of the SFLQ-Br by the general population.

Item	Linguistic Clarity	Appropriateness	Comprehensibility	“No Need to Change It” (%)
#1	0.98	0.98	0.98	90.00
#2	0.93	1.00	0.98	85.00
#2a	0.95	0.99	1.00	95.00
#2b	0.93	0.98	0.99	95.00
#2c	0.92	0.99	0.99	100.00
#2d	0.90	0.96	0.96	95.00
#2e	0.97	0.99	1.00	100.00
#3	0.87	0.94	0.95	75.00
#4	0.83	0.94	0.91	85.00
#5	0.91	0.99	0.95	90.00
#6	0.92	0.99	0.99	100.00
#7	0.90	0.97	0.96	95.00
#8	0.89	0.94	0.95	85.00
#9	0.94	0.99	0.98	100.00
#10	0.92	0.98	0.98	95.00
#11	0.91	0.97	0.97	95.00
#12	0.91	0.99	0.99	100.00
Average CVC	0.92	0.98	0.97	-

**Table 5 foods-11-03968-t005:** Factor loadings, mean, standard deviation, and weighted kappa of each item of the SFLQ-Br.

Item	Factor Loading	Mean ± SD	Weighted Kappa	Weighted Kappa 95% CI	
				Lower	Higher
#1	0.468	3.30 ± 0.83	0.51	0.23	0.77
#2 (average)	0.457	3.43 ± 1.41	0.69	0.46	0.91
#3	0.783	2.44 ± 1.28	0.85	0.73	0.97
#4	0.922	2.28 ± 1.12	0.76	0.55	0.96
#5	0.866	2.36 ± 1.16	0.62	0.37	0.86
#6	0.654	2.63 ± 0.85	0.73	0.53	0.92
#7	0.461	1.93 ± 1.86	0.69	0.46	0.92
#8	0.699	3.74 ± 1.02	0.73	0.57	0.89
#9	0.882	2.66 ± 0.68	0.61	0.42	0.79
#10	0.844	2.68 + 0.73	0.73	0.52	0.93
#11	0.754	2.94 ± 0.61	0.62	0.37	0.86
#12	0.699	2.85 ± 0.79	0.62	0.30	0.95

SD = Standard deviation; 95% CI = 95% confidence interval.

## Data Availability

Data may be provided by the corresponding author upon request.

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
