# Peer review of "Translation, Adaptation, and Validity of the Short Food Literacy Questionnaire for Brazil"

_foods, 2022, doi:10.3390/foods11243968_

Round 1
Reviewer 1 Report
Thank you for submitting the manuscript "Cross-cultural adaptation and content validity of the Short Food Literacy Questionnaire for Brazil" to "Foods". The manuscript is appropriate, the experimental part seems to have been conducted well, but before accepting it for publication I suggest a few points to clarify.
Line#18: It was mentioned that they were food and nutrition experts, but the M&M shows that they were just nutrition experts. Please also confirm with line#126.
Line#154: why?
Line#161: It doesn't seem to me that N:p was previously defined.
Add to the M&M the month and year the experiment was conducted.
Author Response
Thank you for submitting the manuscript "Cross-cultural adaptation and content validity of the Short Food Literacy Questionnaire for Brazil" to "Foods". The manuscript is appropriate, the experimental part seems to have been conducted well, but before accepting it for publication I suggest a few points to clarify.
Response: Thank you for all your suggestions.
Line#18: It was mentioned that they were food and nutrition experts, but the M&M shows that they were just nutrition experts. Please also confirm with line#126.
Response: We agree with the reviewer. They were food and nutrition experts, we included this information in methods section
Line#154: why?
Response: We included a reference for the cutoffs. According to the literature it is suggested all CVCi be higher than 0.80 and the final average (S-CVC) be higher than 0.90. We clarified this issue in the paper.
Line#161: It doesn't seem to me that N:p was previously defined.
Response: We agree with the reviewer. We changed the order of the sentences, for clarity.
Add to the M&M the month and year the experiment was conducted.
Response: We agree with the reviewer. We included this information, as follows:
“The research was conducted from October 2021 to April 2022.”
Reviewer 2 Report
Interesting study to advances proposals for food literacy
Title: I suggest evaliating a change in the title anf instead of food literacy place a validation questionnaire for eating behaviors or similar
Abstract: is suggested to add in the summary the adaptation to the local diet that was made in the questionnaire considering the dietary guidelines for the brasilian population
Introduction.OK
Materials and methods. the process and teh stagesses for the adaptationand validation for the original questionnaire for the brasilian population is adequate and the different statistical test used and the results obtained indicated that this adaptation and validation is pertinent
Results.Ok
Discussion.OK
Conclusions. OK
References.OK
Author Response
Interesting study to advances proposals for food literacy
1) Title: I suggest evaliating a change in the title anf instead of food literacy place a validation questionnaire for eating behaviors or similar
Response: We appreciate your suggestion, however we included in the text the original name of the scale, i.e., Short Food Literacy Questionnaire.
2) Abstract: is suggested to add in the summary the adaptation to the local diet that was made in the questionnaire considering the dietary guidelines for the brasilian population
Response: We agree with the reviewer. We included the following sentence in the abstract:
“The SFLQ has been adapted for the Brazilian context and guidelines.”
3) Introduction.OK
Response: Thank you for all the suggestions and support.
4) Materials and methods. the process and the stages for the adaptation and validation for the original questionnaire for the brasilian population is adequate and the different statistical test used and the results obtained indicated that this adaptation and validation is pertinent
Response: Thank you for your comments.
5) Results.Ok; Discussion.OK; Conclusions. OK; References.OK
Response: Thank you for all the suggestions and support.
Reviewer 3 Report
Dear Authors,
I believe the topic is of interest and relevance for the food scientists and nutritionist. The overall concept of the paper is well considered but there are some improvements needed before considering it for further processing.
Title
Cross-cultural adaptation and content validity of the Short Food Literacy Questionnaire for Brazil
I am not truly convinced that starting the title with the term “cross-cultural adaptation” is the most appropriate as it goes beyond the real scope of the study.
Material and methods
Mention origin of JASP software and indicate it is open-source tool supported by the University of Amsterdam
Mention why you decided to use the EN version if the original scale was developed in German.
What about backtranslation process? Did you perform it? If not discuss it within the limitations.
I miss the information on the correlations between individual items of the scale. It can be added as a supplementary file.
Author Response
Dear Authors,
I believe the topic is of interest and relevance for the food scientists and nutritionist. The overall concept of the paper is well considered but there are some improvements needed before considering it for further processing.
Response: Dear reviewer, thank you for all your suggestions.
1) Title: Cross-cultural adaptation and content validity of the Short Food Literacy Questionnaire for Brazil. I am not truly convinced that starting the title with the term “cross-cultural adaptation” is the most appropriate as it goes beyond the real scope of the study.
Response: We agree with the reviewer. The title was changed, as follows:
“Translation, adaptation and validity of the Short Food Literacy Questionnaire for Brazil”
2) Material and methods- Mention origin of JASP software and indicate it is open-source tool supported by the University of Amsterdam
Response: We agree with the reviewer. We included this information, as follows:
“The CFA and internal consistency were conducted using JASP 0.16.4 (JASP Team - Netherlands - Open-source tool supported by the University of Amsterdam).”
3) Mention why you decided to use the EN version if the original scale was developed in German.
Response: While the original version was in German, the authors have published the EN version of the questionnaire. The German version can only be found on their University's repository. We have included this information in the method section.
4) What about backtranslation process? Did you perform it? If not discuss it within the limitations.
Response: We agree with the reviewer. We included two sentences in the limitations, as follows:
“Another limitation was the lack of back-translation. While many studies use only the forward-translation, the back-translation might be helpful for minor adjustments.”
5) I miss the information on the correlations between individual items of the scale. It can be added as a supplementary file.
Response: We agree with the reviewer. We included information about the correlations in the text and added a table as supplementary file.